# Three Birds, One Excipient: Development of an Improved pH, Isotonic, and Buffered Ketamine Formulation for Subcutaneous Injection

**DOI:** 10.3390/pharmaceutics14030556

**Published:** 2022-03-03

**Authors:** Jason Wallach, James Gamrat, Rebekah Jauhola-Straight, Jeffrey Becker, Thomas Eckrich

**Affiliations:** 1Department of Pharmaceutical Sciences, University of the Sciences, Philadelphia, PA 19104, USA; jgamrat@usciences.edu (J.G.); rjauhola-straight@usciences.edu (R.J.-S.); 2Bexson Biomedical Inc., Santa Barbara, CA 93108, USA; jeffrey@bexsonbiomedical.com (J.B.); eckrichconsulting@gmail.com (T.E.)

**Keywords:** ketamine, captisol, cyclodextrin, ketamine formulation, osmolality, subcutaneous injection

## Abstract

Subcutaneous (SC) ketamine has been found to be effective in pain management, though reports of injection site irritation and sterile abscesses exist with currently available ketamine HCl formulations. Such adverse SC reactions are commonly associated with low pH, high osmolality and/or high injection volumes. An optimal SC formulation of ketamine would thus have a pH and osmolality close to physiological levels, without compromising on concentration and, thus, injection volume. Such a formulation should also be buffered to maintain the pH at the acceptable level for extended time periods. As many of these physicochemical properties are interrelated, achieving these aims represented a significant challenge in formulation development. We describe the development of a novel Captisol^®^-based formulation strategy to achieve an elevated pH, isosmotic and buffered formulation of ketamine (hence, three birds, one excipient) without compromising on concentration. This strategy has the potential to be readily adapted to other amine-based APIs.

## 1. Introduction

Cyclodextrins are cyclic oligomers composed of varying numbers of glucopyranose monomers. This cyclic structure confers unique and remarkable properties to cyclodextrins. Properties include the ability to complex and solubilize various organic small molecules. As such, various cyclodextrins have found use in consumer, industrial and medicinal products [1]. In medicine, cyclodextrins have become common excipients of pharmaceutical formulations. Their utility comes from the fact that cyclodextrins form inclusion complexes with small organic molecules. The resulting complex can be referred to as a supramolecular guest–host interaction between the organic molecule (guest) and cyclodextrin (host) [2]. This complex is thermodynamically favored by a negative Gibbs free energy change (ΔG), and is stabilized by hydrophobic and/or electrostatic interactions between the guest and the inner hydrophobic core of the cyclodextrin host. In contrast to the inner core, the outer ring of a cyclodextrin is highly polar, conferring high aqueous solubility to the cyclodextrin and the resulting complex. In this way, cyclodextrins are able to solubilize organic molecules with otherwise poor aqueous solubility [2].

One of the most common cyclodextrins used in medicine is *β*-cyclodextrin (having seven glucopyranose units) and its substituted derivatives 2-hydroxypropyl-*β*-cyclodextrin (HP*β*CD) and sulfobutylether-*β-*cyclodextrin (SBE*β*CD) [2]. The latter differ by the addition of polar side chains that enhance the aqueous solubility of the cyclodextrins. HP*β*CD and SBE*β*CD can further vary depending on the extent of side chain modification [2,3]. A common commercial form of SBE*β*CD is Captisol^®^, which contains ~6.5 sulfobutylether side chains per *β-*cyclodextrin ring [4]. Captisol^®^ has been used to improve the solubility and stability of small organic molecule active pharmaceutical ingredients for use in intravenous (IV) and other parenteral injection formulations [4]. At least nine FDA-approved drug products contain Captisol^®^ [5]. For instance, the FDA-approved antiarrhythmic, Nexterone, contains amiodarone HCl complexed with Captisol^®^ to allow for a bolus IV injection while reducing the adverse hypotensive effects observed with previous formulations [6]. Captisol^®^ is used as a suitable replacement for co-solvents such as propylene glycol, as in the case of Evomela, an FDA-approved melphalan injectable for cancer treatment [7]. Orally administered antipsychotics ziprasidone and aripiprazole have been formulated with Captisol^®^ to produce formulations suitable for intramuscular (IM) injection [2]. While there are diverse applications for this excipient, most research and existing applications involve complexation of small organic molecules with Captisol^®^ to improve solubility.

Common issues associated with parenteral injections include pain, irritation and/or ulceration at the site of injection. This is especially an issue with subcutaneous (SC) and intramuscular (IM) injection [8]. Such adverse responses are believed to be caused by several factors. One of the major factors that appears to dictate these adverse effects is the osmolality of the injection solution [9]. Osmolality is a measure of dissolved solids in a solution. A clinically optimal solution for IM and SC injection should be isotonic and isosmotic (~300 mOsm) and it is recommended to not exceed 600 mOsm [8,9]. SC injection of hypertonic saline solutions has been reported to produce injection site pain and reactions in humans that increase proportionally to the solution’s osmolality [8]. Additionally, hypertonic formulations for SC injection increase the possibility of tissue and cell damage [8]. In addition to osmolality, the pH of the solution should ideally be close to physiological pH to reduce irritation and tissue damage [9]. Finally, injection volume and viscosity are additional factors that can influence SC and IM injection site pain and reactions [8].

Ketamine is a small molecule N-methyl-D-aspartate receptor (NMDAR) antagonist that has traditionally been used in medicine as a dissociative anesthetic [10,11]. The S-enantiomer has recently been approved for treatment-resistant depression. NMDAR antagonists have emerged as promising agents for various types of pain including chronic pain [12] and neuropathic pain [13]. Intravenous ketamine has previously been used to treat acute pain at subanesthetic doses with positive results in patients suffering from postoperative pain [14,15]. Guidance on the use of intravenous ketamine infusions in acute and chronic pain has recently been published [11,16]. Ketamine is generally administered via parenteral routes to overcome its poor oral bioavailability (8–24%) [17]. Likewise, ketamine has rapid systemic clearance due to its short elimination half-life, thus often requiring a continuous infusion to maintain efficacious plasma levels [17]. While the intravenous route is effective and has high bioavailability, it is a limiting route due to the difficulty involved in its implementation. SC administration offers a useful alternative to IV ketamine in that it has high bioavailability (75–95%) and ease of both bolus and continuous administration [16]. It can thus be a safer and more convenient alternative to IV. While SC ketamine is effective in pain management, reports of injection site irritation and sterile abscesses exist with ketamine HCl formulations [18,19]. We hypothesize that these adverse reactions are due, at least in part, to a low pH and/or high osmolality characteristic of commercial ketamine HCl formulations. An optimal SC formulation of ketamine would thus be closer to physiological pH and osmolality without compromising on concentration and thus volume. A buffered solution is also necessary in order to maintain pH for extended time periods. Since many of these physicochemical properties are interrelated, achieving these aims represents a significant challenge in formulation development. We describe the development of a novel Captisol^®^-based formulation strategy (Figure 1) to achieve an elevated pH, isosmotic and buffered formulation of ketamine without compromising on concentration. Finally, the approach described has the potential to be adapted for other amine-based APIs.

## 2. Materials and Methods 

### 2.1. General Procedures

Masses used to prepare stock solutions of Captisol^®^ were corrected for moisture content (Karl Fischer titration, described below). All aqueous stock solutions were prepared using HPLC-grade water from Sigma-Aldrich (St. Louis, MO, USA). Equivalence points were determined by titration with 0.5 M NaOH (Sigma Aldrich, St. Louis, MO, USA). Linear and non-linear regressions were performed in OriginPro 2020 (OriginLab Corporation, Northampton, MA, USA).

### 2.2. Chemicals and Reagents

Solvents (water, acetonitrile, and isopropanol) were HPLC Plus grade and obtained from Sigma Aldrich (St. Louis, MO, USA). The following chemicals were obtained and used without further purification: ammonium formate (≥99.9%, Sigma Aldrich, St. Louis, MO, USA), USP grade ketamine HCl (Spectrum, Gardena, GA, USA), procaine HCl (≥97%, Sigma Aldrich, St. Louis, MO, USA), formic acid (≥98%, Sigma Aldrich, St. Louis, MO, USA), ammonium acetate (LiChropur, HPLC grade, Supelco, Bellefonte, PA, USA), Amberlite IRC120 H hydrogen form resin (Sigma Aldrich, St. Louis, MO, USA), ethanol (200 proof, Pharmco, Greenfield Global, Toronto, ON, Canada), chloroform-d (≥99.8% atom D, Sigma Aldrich, St. Louis, MO, USA), sodium hydroxide (≥97%, Sigma Aldrich, St. Louis, MO, USA), potassium hydroxide (≥85%, Sigma Aldrich, St. Louis, MO, USA), sodium chloride (Certified ACS crystalline, Fisher Scientific, Fair Lawn, NJ, USA), sodium potassium tartrate tetrahydrate (Sigma Aldrich, St. Louis, MO, USA), (+)-tartaric acid (99%, Alfa Aesar, Haverhill, MA, USA), cupric sulfate (Drug & Chemical Co, Inc. Irvington, NJ, USA), glucose (Sigma Aldrich, St. Louis, MO, USA), Benzethonium chloride (≥99%, Sigma Aldrich, St. Louis, MO, USA), Captisol^®^ (Lot # NC-04A-180185, Cydex Pharmaceuticals, Lawrence, KS, USA), propylene glycol (Florida Laboratories Inc., Fort Lauderdale, FL, USA), β-cyclodextrin (99.5+%, ChemCenter, La Jolla, CA, USA), and 2-hydroxypropyl-β-cyclodextrin (97+%, Acros Organics, Fair Lawn, NJ, USA). The following commercial ketamine HCl samples were obtained: Mylan (Lot #190524, 50 mg/mL, Canonsburg, PA, USA), Ketalar (Lot #338718, 50 mg/mL), Hospira (Lot #11215DD, 100 mg/mL, Lake Forrest, IL, USA), and WestWard (Lot #2005078.1, 100 mg/mL, Eatontown, NJ, USA).

### 2.3. Instrumentation

#### 2.3.1. pH and Mass Measurements

pH readings were obtained by an Orion 3 star (Thermo Scientific, Waltham, MA, USA) pH meter equipped with either a Thermo pH electrode (9142BN) or an Orion 8103BNUWP Ross Ultra Semi-micro pH probe (Thermo Scientific, USA) filled with 3M KCl ROSS Orion filling solution (Thermo Scientific, USA). An Ohaus ADVENTURER AX124 analytical balance (Ohaus, Parsippany, NJ, USA) was used for mass measurements. Samples were weighed on 3 × 3 inch low-nitrogen-weighing Fisherbrand paper (Fisherbrand, Pittsburgh, PA, USA). In general, a mass of at least 5 mg was weighed for all samples to minimize error. Both the balance and pH meter were calibrated directly prior to use. Balance calibration was confirmed with a 5 mg standard weight (Troemner, Thorofare, NJ, USA) with 5 ± 0.1 mg cutoff.

#### 2.3.2. Nuclear Magnetic Resonance Spectroscopy 

^1^H and ^13^C NMR spectra data were obtained on a Bruker Avance III with PA BBO 400S1 BBF-H-D-05 Z plus probe (Bruker Corporation, Billerica, MA, USA). Samples were prepared at a concentration of ~20 mg/mL in CDCl_3_ (Sigma-Aldrich, St. Louis, MO, USA). Chemical shifts are reported in parts per million (ppm), with the reference shift set to 7.26 ppm (residual CHCl_3_) for ^1^H and solvent signal for ^13^C (δ = 77.16 ppm).

#### 2.3.3. GC/MS

GC/MS analyses were performed on a Thermo Scientific Trace 1300 Gas Chromatograph coupled to a Thermo Scientific ISQ QD Single Quadrupole Mass Spectrometer. A Thermo Scientific TraceGold TG-5MS GC Column (30 m × 0.25 mm × 0.25 µm) was utilized for separation. Samples were prepared at a concentration of 1 mg/mL in ethyl acetate. Ionization was achieved by electron impact (EI). Data were analyzed using Thermo Xcalibur ^TM^ Software (version 3.1.66.10, Thermo Scientific, Waltham, MA, USA). Transfer line and ion source were set to 210 °C and 200 °C, respectively. The starting temperature of the oven was 100 °C and held for 1 min. Temperature was increased at a rate of 8 °C/min until reaching 220 °C, at which point it was held for 4 min. The total time of the run was 20 min.

#### 2.3.4. Moisture Readings

Moisture readings were obtained on a Mettler Toledo C20 Coulometric Karl Fischer titrator (Mettler Toledo, Columbus, OH, USA) charged with HYDRANAL™-Coulomat AG reagent from Honeywell Fluka™. A sample of Aquastar Water standards (0.1% moisture) from EMD Millipore and/or HYDRANAL™ (10 mg water/g solution) from Honeywell Fluka™ was run each time as a positive control. Measurements were made on masses > 70 mg.

#### 2.3.5. Osmolality

Osmolality readings were obtained on an Advanced Instruments Advanced™ Micro Osmometer Model 3300 (Advanced Instruments, Inc. Norwood, MA, USA). A saline solution (0.9%) (AddiPak, Teleflex Inc., Wayne, PA, USA) was run as a positive control. Each sample was calculated as the mean from back-to-back triplicate measurements. When replicate batches were measured, data are reported as mean +/− SEM.

#### 2.3.6. HPLC

HPLC analyses were performed on an Agilent 1260 Infinity system that includes a 1260 quaternary pump VL, a 1260 ALS autosampler, a 1260 Thermostatted Column Compartment, and a 1200 DAD Multiple Wavelength Detector (Agilent Technologies, Santa Clara, CA, USA). The detection wavelength was set at 220 nm. Separation for KetCap solutions was achieved using a Zorbax Eclipse Plus-C18 analytical column (5 µm, 4.6 × 150 mm) from Agilent (Agilent Technologies, Santa Clara, CA, USA). Mobile phase A consisted of 10 mM aqueous ammonium formate buffer titrated to pH 4.5 and mobile phase B consisted of acetonitrile. The injection volume of samples was 40 µL, flow rate was 1.0 mL/min, and the column temperature was set at 40 °C. All samples were injected in duplicate with a wash of the injector (30:70 A:B) between runs. Run time was 10 min with a mobile phase ratio (isocratic) of 70% A and 30% B. The HPLC method was adapted from literature methods used previously for ketamine and related basic amines [20,21]. Control experiments were performed to validate the method with respect to accuracy, precision, linearity and to establish that added Captisol^®^ did not alter the peak area of ketamine under the assay conditions. The chiral HPLC method used was adapted from the literature [22]. Separation for chiral HPLC was achieved on the same system using a Supelco Analytical Chiral-AGP™ column (5 µm, 3.0 × 150 mm) (Supelco Inc., Bellefonte, PA, USA). Mobile phase A consisted of 10 mM aqueous ammonium acetate buffer titrated to pH 7.6 and mobile phase B consisted of isopropanol. The injection volume of samples was 10 µL, flow rate was 0.4 mL/min, and the column temperature was set at 25 °C. Samples were injected in duplicate with an injector wash (95:5 A:B) between runs. Run time was 20 min with a mobile phase ratio (isocratic) of 95% A and 5% B. Chromatograms were analyzed using the Agilent ChemStation Software (Agilent Technologies, Santa Clara, CA, USA).

### 2.4. Fehling’s Colorimetric Assay for Stability of Captisol^®^

Captisol^®^ stability tests were performed by use of a qualitative colorimetric assay using Fehling’s reagent. Stock solutions were prepared as follows: Solution A—3.5% CuSO_4_∗5H_2_O in deionized H_2_O; Solution B—12.5% potassium hydroxide and 17.3% sodium potassium tartrate in deionized H_2_O [23]. Fehling’s reagent was prepared by adding equal volumes of Solution A and Solution B to make a vibrant blue solution directly prior to use. Fresh reagent was prepared for each experiment. The stability test was performed by adding Fehling’s reagent (2 mL) to a solution of analyte (2 mL) in a borosilicate glass test tube and heating at 60 °C in a water bath for 2 min. The formation of an orange/red solid precipitate indicates a positive result for the presence of reducing sugars and no change in the blue color of the solution indicated a negative result for the presence of reducing sugars. The process is summarized in Appendix A. 

To test the stability of CapAcid prepared from 15% and 20% Captisol^®^, CapAcid was prepared via ion exchange as described and the eluent aliquoted into test tubes. The samples were tested at timepoints 0, 24, 48, and 72 h along with negative controls (40% Capitsol and HPLC H_2_O) and positive controls (10 mg/mL glucose and 2 mg/mL glucose). Each set of samples was analyzed in duplicate, with results assessed at 5 and 20 min. Experiments were carried out in triplicate. The concentration of CapAcid in the solution was determined by mass following lyophilization (corrected for moisture by KF titration).

### 2.5. Titration Method

Titrations were performed by adding a solution of analyte into a 50 mL borosilicate glass beaker containing a Teflon-coated magnetic stir bar. With stirring, a solution of 0.5 M NaOH was then added in small portions via micropipette to the analyte and the pH was read after each addition. Data were plotted in OriginPro 2020 (OriginLab Corporation, Northampton, MA, USA) and a non-linear regression analysis was performed for a sigmoidal curve fit. The equivalence point was determined from the regression analysis and compared to the calculated equivalence point. Citric acid (0.3 M) and hydrochloric acid (0.48 M) were used as controls to assess the accuracy of this method of titration (and were within 95–105% of the theoretical equivalence point). This was deemed sufficient for titration of Captisol^®^ acid (CapAcid).

### 2.6. Ketamine Freebase Captisol^®^ Phase Solubility

The solubility of ketamine freebase in Captisol^®^ was determined by a phase solubility experiment in a modification of an experiment described by Ligand [24]. Briefly, a 40% (*w/w*) solution of Captisol^®^ was prepared and serially diluted to the remaining concentrations (2.5%, 5%, 10% and 20%) to a final volume of 1.0 mL per test tube (borosilicate glass). To each test tube was added a Teflon-coated magnetic stir bar, and solid ketamine free base as follows: 100 mg ketamine freebase in 40% (*w/w*) Captisol^®^ solution, 50 mg ketamine freebase in the remaining solutions. The test tubes were carefully sealed with parafilm, placed in a large beaker, protected from light with aluminum foil, and allowed to stir (magnetic stirring) at room temperature for 48 h. Samples were then prepared for HPLC analysis, which consisted of filtration through a 0.45 µM nylon filter. An aliquot from each sample was then diluted in 7:3 formate:acetonitrile buffer with 20 µg/mL procaine HCl to an estimated target range of 20–40 µg/mL. Samples were diluted in duplicate and each sample was analyzed by HPLC with procaine as an internal standard to obtain the ratio of ketamine area/procaine area for analysis. The quantity of ketamine solubilized in each concentration of Captisol^®^ was determined by linear regression analysis of molar [Ketamine] vs. [Captisol^®^] in Origin. Individual samples were run in duplicate and the overall experiment was run in triplicate. Complexation constant (K_1:1_) was calculated as the linear regression line slope/(intrinsic solubility/1-slope). The intrinsic solubility (S_0_), the solubility of ketamine in water, was determined experimentally to be 0.88 mg/mL. Complexation efficiency (CE) was calculated as S_0_ × K_1:1_ [3]. 

### 2.7. Preparation of Captisol^®^ Acid (CapAcid)

Amberlite IR120 Hydrogen form resin (62 g, 4.4 meq/g, 20 equivalents) was soaked in HPLC-grade water (250 mL) for 5 min and loaded onto a borosilicate glass column (30 mm diameter) containing a washed (HPLC-grade water) cotton plug. The resin was washed with 2 column volumes of HPLC-grade water and dried with air pressure for 10 min. A solution of 15% Captisol^®^ (30 mL) was then added to the column and air pressure used to elute the solution into a tared, aluminum foil-covered beaker. Care was taken to avoid interaction between the foil and the solution. The resin was then washed with HPLC-grade water (30 mL), which was collected separately. Eluents were lyophilized using an SP Scientific Wizard 2.0 lyophilizer at 30 mmHg for 48 h. The resulting white solids were crushed with a glass stir rod (contact with metals was avoided out of caution), weighed, and water content was determined by Karl Fischer titration. The resulting white, shiny solids (3.30–4.04 g, 78.5–96% recovery, 4.40–7.80% moisture, N = 5) were stored in an aluminum-foil-covered (to keep dark) and parafilm-sealed 20 mL scintillation vial at −20 °C. When stored in this manner, no visible discoloration of the material occurred even after many months.

### 2.8. Synthesis of Ketamine Freebase

Ketamine hydrochloride (3.0 g, 10.9 mmol) was dissolved in nanopore H_2_O (100 mL). Then, 2 M NaOH was added with stirring until a cloudy white precipitate formed. An excess of this base solution was added such that additional quantities did not cause further precipitation. The solution was stirred, allowed to stand for 20 min at 0 °C, after which the precipitate was collected by vacuum filtration, washed with 300 mL of HPLC-grade water and dried under vacuum. Recovery of the precipitated freebase was ~2.0 g. The remaining material was obtained by extraction of the basic aqueous solution with ethyl acetate (3 × 50 mL), followed by washing the pooled organic extractions with brine (10 mL) and evaporating under vacuum to provide residual solids. The white solids were combined to provide 2.36 g of ketamine freebase (95.7% recovery). Subsequent conversions using 2.5 M equivalents of NaOH and cooling on and ice-bath for 20 min led to near quantitative recovery on filtering. Purity and identity of the freebase were confirmed by ^1^H and ^13^C NMR, HPLC, and GC-MS. Moisture was determined by KF titration (<1%). ^1^H NMR (400 MHz, CDCl_3_) δ 7.54 (dd, *J* = 7.8, 1.6 Hz, 1H, H_6′_), 7.37 (dd, *J* = 7.8, 1.4 Hz, 1H, H_3′_), 7.31 (td, *J* = 7.6, 1.4, 1H, H_5′_), 7.23 (td, *J* = 7.7, 1.6, 1H, H_4′_), 2.79-2.74 (m, 1H, H_6_), 2.51-2.45 (m, 2H, H_3_), 2.16 (br s, 1H, NH), 2.10 (s, 3H, NCH_3_), 2.05–1.94 (m, 1H, H_4_), 1.92–1.81 (m, 1H, H_4_), 1.81–1.68 (overlap m, 3H, H_5_ and H_6_). ^13^C NMR (100 MHz, CDCl_3_) 209.10 (C_2_), 137.63 (C_1′_), 133.98 (C_2′_), 131.38 (C_3′_), 129.62 (C_6′_), 128.93 (C_4′_), 126.82 (C_5′_), 70.39 (C_1_), 39.68 (C_3_), 38.77 (C_6_), 29.23 (NCH_3_), 28.26 (C_4_), 21.98 (C_5_). mp: 92.0–93.1 °C, Lit: 92.5 °C [25]. Spectra are presented in Appendix A.

### 2.9. Resolution of S-Ketamine 

Modification of the method reported by Steiner et al. [26] was used. R,S-ketamine free base (500 mg, 2.10 mmol) was dissolved in acetone (6.1 mL) and l-(+)-tartaric acid (315 mg, 2.10 mmol) was added. Deionized H_2_O (400 µL) was added and the mixture was heated to boiling. The solution was then cooled to rt and left for several days. No solids had formed and the solution was then placed to 4 °C. After several hours, a dense cluster of long thin white crystalline needles formed. The solids were collected (400 mg) by decanting, washed with acetone and dried. The resulting solids were recrystallized once from ~10 mL acetone:water (8:1) at room temperature, to form needle-like crystals (crop 1 = 270 mg). Specific rotation α = +74.47 (water, 25 °C). Literature α = +68.9 (water) [26]. To obtain the freebase, the S-ketamine (+)-tartrate salt was dissolved in deionized H_2_O and basified with excess 2M NaOH to form a white solid precipitate. The solids were collected by vacuum filtration, washed with HPLC-grade water (~100 mL) and dried in vacuo to afford S-ketamine freebase (100 mg, 0.42 mmol, 40% yield) as a white, fluffy solid. The freebase could be recrystallized from a small volume of boiling cyclohexanes. Identity and purity were determined using chiral HPLC (Appendix A). mp: 118.3–120.2 °C, Lit: 120 °C [27]. (S)-ketamine freebase [a]_D_: −57.19° (c = 1/EtOH, 26 °C), Lit: −55.8°, −56.35° [26]. Reference compound: L-(+)-TA = 14.79. HPLC % purity: 99.2%. 100% ee.

#### Resolution of R-Ketamine

The supernatant from the first crystallization of Ketamine tartrate described in the (+)-ketamine section above was collected and evaporated under vacuum. The resulting white crystalline solids were dissolved in ~5 mL boiling isopropanol, covered and left undisturbed. After sitting for several days, the large transparent cubic salt-like crystalline clusters were then collected by decanting and washing sparingly with isopropanol and air dried (350 mg white-transparent crystalline solids). These solids were dissolved in 10 mL deionized H_2_O. KOH pellets were added until basic pH (>11) and extracted with 4 × 50 mL ethyl acetate, washed with 10 mL brine, extractions pooled, dried over anhydrous Na_2_SO_4_, and evaporated under a stream of hot air to give 160 mg (0.67 mmol, 67% yield) of a colorless oil. This oil crystallized to a white solid on sitting and was recrystallized from a small quantity of boiling cyclohexane, to give small transparent cubic crystals. HCl salt was prepared by dissolving in 20 mL ethanol, titrating to pH ≥ 1 with concentrated HCl. The solvent was evaporated; ethanol was repeatedly added and evaporated until all residual HCl and H_2_O appeared gone. The resulting white crystalline solids were washed with Et_2_O (2 × 5 mL), dried and recrystallized by dissolving in 5 mL boiling ethanol and adding 15 mL Et_2_O to obtain white needles on standing. Identity and purity were determined using chiral HPLC (Appendix A). mp: 119.7–120.1 °C, (R)-ketamine HCl [a]_D_: −90.42° (c = 1/EtOH, 26 °C), (R)-ketamine freebase [a]_D_: +53.75° (c = 1/EtOH, 26 °C), Reference compound: L-(+)-TA = 14.79. HPLC % purity: 99.3%. 100% ee.

### 2.10. Preparation of Ketamine Captisol^®^ Formulations

#### 2.10.1. Preparation of BB105 (96 mg/mL Ketamine Freebase)

CapAcid (698 mg, moisture corrected 753 mg) was dissolved in HPLC-grade water (4 mL) in a borosilicate glass test tube containing a Teflon-coated magnetic stir bar. The solution was stirred vigorously and ketamine freebase (480 mg) was added in ~20–40 mg portions; each portion was allowed to completely dissolve before the next was added. The solid was agitated as necessary to facilitate dissolving. When the additions were complete, benzethonium chloride (5 mg, to give a final concentration of 0.1%) was added and 2 M sodium hydroxide was added in small portions via a pipettor until the pH was raised to 5.5 (initial starting pH of the solution was 1.39). ~155 µL of 2 M NaOH was generally necessary. The final pH of different batches ranged from pH 5.50 to 5.54. When complete, the solution was carefully transferred to a volumetric flask and the volume was brought up to 5 mL. The solution was syringe-filtered with a 0.45 µM nylon filter and stored in a parafilm-sealed borosilicate glass test tube under ambient air and wrapped in aluminum foil. pH was confirmed to be stable after the final dilution. Osmolality readings were obtained immediately after preparation. 

#### 2.10.2. Preparation of BB106 (70 mg/mL Ketamine Freebase) 

BB106 was prepared by the same method as BB105 on a 10 mL scale with the following: CapAcid (5.80% moisture, 1.0 g active, 1.066 g moisture corrected), racemic ketamine free base (700 mg), benzethonium chloride (10.0 mg, to give a final concentration of 0.1%). Titrations were performed with small portions of 2M NaOH (~160 µL total) to reach the target pH of 5.5. The final pH ranged from pH 5.50 to 5.57. Osmolality readings were obtained immediately after preparation. 

A portion of BB106 (3 mL) was lyophilized to produce a white solid plug that was ground to a stable white powder (KetCap, 510.7 mg, ~quantitative mass recovery). Moisture of the sample was determined by KF titration (4.0%). This resulting KetCap powder did not visibly discolor and only appeared hygroscopic under especially humid ambient conditions. The powder was readily reconstituted in water and pH of the reconstituted formulation was stable (pH 5.50).

#### 2.10.3. Preparation of BB107 (70 mg/mL S-Ketamine Freebase)

BB107 was prepared by the same method as BB105 on a 5 mL scale with the following: CapAcid (5.95% moisture, 500 mg active, 531 mg moisture corrected), S-ketamine free base (350 mg), benzethonium chloride (5.0 mg, to give a final concentration of 0.1%). Titrations were performed with small portions of 2M NaOH (80 µL total) to reach the target pH of 5.5. The final pH ranged from pH 5.50 to 5.57. Osmolality readings were obtained immediately after preparation. pH = 5.50.

#### 2.10.4. Preparation of BB108 (70 mg/mL R-Ketamine Freebase)

Prepared by the same procedure as BB107 on a 5 mL scale using R-ketamine. Final pH = 5.52.

#### 2.10.5. Preparation of Ketamine-HCl-Captisol^®^ Formulation (70 mg/mL Ketamine HCl FB Eq)

Captisol^®^ (5.53% moisture, 214.8 mg active, 227.4 mg moisture-corrected) was dissolved in HPLC water (1.0 mL) in a borosilicate glass test tube containing a Teflon-coated magnetic stir bar. The solution was stirred using a magnetic stir plate and racemic ketamine HCl (161.4 mg) was carefully added in portions via a metal spatula (~50–70 mg portions). Following the addition of the ketamine, benzethonium chloride (2.0 mg, to give a final concentration of 0.1%) was added and the solution was then titrated slowly by pipetting small portions of 2M NaOH (~15 µL total) to reach the target pH of 5.5. The solution was carefully transferred and adjusted with HPLC-grade H_2_O to a final volume of 2.0 mL in a volumetric flask. The solution was then filtered through a 0.45 µM nylon syringe filter, and stored in a sealed vial at room temperature in the dark (aluminum foil). Osmolality readings were obtained immediately after preparation. pH = 5.49.

#### 2.10.6. Preparation of Solid KetCap Stoichiometric Salt

CapAcid (5.68% moisture, 650 mg active, 689.2 mg moisture corrected) was dissolved in HPLC water (5.0 mL) in a borosilicate glass test tube containing a Teflon-coated magnetic stir bar. The solution was stirred using magnetic stirring and racemic ketamine freebase (500 mg) was carefully added in one portion and stirred until a homogenous solution formed. The solution was then filtered through a 0.45 µM nylon syringe filter and subjected to lyophilization to afford KetCap as a white, fluffy solid (1.14 g, 99.1% recovery). The sample was stored at room temperature in a scintillation vial, sealed with parafilm, and covered with aluminum foil. When stored in this manner, the material remained a free flowing solid for many months. 

#### 2.10.7. Preparation of BB106 from CapAcid Solution

Amberlite IR120 Hydrogen form resin (81 g, 4.4 meq/g, 20 equivalents) was soaked in HPLC-grade water (250 mL) for 5 min and loaded into a borosilicate glass column (30 mm diameter) containing a washed (HPLC-grade water) plug of cotton. Once loaded, the resin was washed with 2 column volumes of HPLC-grade water and freed of excess water with positive air flow for 10 min. A solution of 20% Captisol (30 mL) was then added to the column and air pressure used to elute the resulting solution into a tared, aluminum-foil-covered beaker. Next, 3 × 2 mL portions of the eluent were transferred to tared scintillation vials and lyophilized for 24 h to determine the concentration of CapAcid by mass (13.99% CapAcid). The remaining solution was sealed with parafilm and stored at 4 °C until use. CapAcid solution (3.57 mL) was added to a borosilicate glass test tube containing a Teflon-coated stir bar, and ketamine freebase (350 mg) was added to form a clear homogenous solution. Benzethonium chloride (5 µL of a 1% solution, to give a final concentration of 0.1%) was added and the pH was adjusted to 5.5 with 2M NaOH (52 µL). The solution was then carefully transferred to a 5 mL volumetric flask and diluted to the target volume with HPLC-grade H_2_O. The final pH of the solution was recorded and osmolality was immediately determined and compared with 0.9% saline standard. Final pH: 5.55; Osmolality: 269 mOsm/kg.

### 2.11. Analysis of Ketamine Captisol^®^ Formulations

#### 2.11.1. Concentration of KetCap and Commercial Ketamine HCl Samples

##### HPLC Sample Preparation and Analysis

For each individual experiment, a ketamine standard curve was generated and used to calculate unknown concentrations. Standard solutions of ketamine were made in running buffer (3:7 acetonitrile (Mobile phase B) to aqueous 10 mM ammonium formate pH 4.5 (mobile phase A), with 20 µg/mL procaine as an internal standard. The ketamine standards were prepared from ketamine HCl in buffer at 1 mg/mL (freebase equivalents, FB equiv) stock solution using serial dilution to the following concentrations: 40, 30, 20, 10, and 5 µg/mL. BB105 (96 mg/mL ketamine), BB106 (70 mg/mL ketamine), Ketalar Ketamine HCl (50 mg/mL), Mylan Ketamine HCl (50 mg/mL), Hospira Ketamine HCl (100 mg/mL), and West-Ward Ketamine HCl (100 mg/mL) samples were diluted to the target concentration of 20 µg/mL in running buffer. The lowest volume pipetted at any point was 50 µL to reduce error. Samples were diluted in duplicate, each sample was run in duplicate, and the overall experiment was performed in triplicate. The standard curve was plotted in Origin using linear regression analysis of the concentration against the ketamine:procaine peak area (absorbance at 220 nm). Ten-month stability of BB106 was determined using a variation of the HPLC method with higher standard concentrations 400, 300, 200, 100, and 50 µg/mL and 2:8 acetonitrile:ammonium formate buffer (10 mM, pH 4.5), which was linear but gave a more robust signal for quantification. Concentrations of sample stock solutions were determined using linear regression using the ketamine calibration curve followed by calculation of starting concentration based on dilution factors.

#### 2.11.2. Enantiomeric Purity of S-Ketamine and R-Ketamine Freebase

The enantiomeric purities of S-Ketamine and R-Ketamine freebase were determined by chiral HPLC. The respective freebases (3 mg) were converted to the HCl salt by dissolving in EtOH (3 mL) and adding a molar equivalent of 1M HCl (14 µL). Solvent and excess HCl were evaporated and the salts were subjected to 4 cycles of reconstituting in EtOH (3 mL) and evaporation. The samples were dissolved in 10 mM ammonium acetate pH 7.6 to a final concentration of 1 mg/mL (FB equiv) and injected onto the chiral column. Enantiomeric purity was determined from the relative peak areas of the respective isomers at 220 nm. Racemic ketamine HCl was run as a control at 1 mg/mL (FB equiv).

#### 2.11.3. Density

Triplicate aliquots of BB106 (1.0 mL) were transferred to a tared 20 mL borosilicate glass scintillation vial and weighed to measure the density of the BB106 formulation. This process was performed on two separate batches of BB106 (N = 3) to generate a mean density.

#### 2.11.4. Buffering Capacity: Forced Precipitation of Ketamine HCl and Formulations with 1 M NaOH

Solutions of BB106 (70 mg/mL ketamine) and ketamine HCl (70 mg/mL FB equiv) with and without additives (10% propylene glycol, 10% HPβCD, and 10% Captisol^®^) were prepared and titrated to a starting pH of 5.5 ± 0.01 with 2 M NaOH. Solutions were then stirred (magnetic stirring with a Teflon-coated magnetic stir bar) in a borosilicate glass test tube. Then, 2 M NaOH (1 µL per addition) was added via micropipette while monitoring pH. The solutions were stirred for 30 s after each addition of NaOH. Addition was continued until a visible precipitate (presumed ketamine freebase) formed. Each sample was analyzed once and the experiment was repeated three times (N = 3) to generate a mean ± SEM. 

#### 2.11.5. Buffering Capacity: Titration of Ketamine HCl and Captisol^®^ Formulations with 1 M HCl

Solutions of KetCap, ketamine HCl, and ketamine-HCl-Captisol^®^ (100 mg/mL Ketamine FB equiv, pH 5.5 ± 0.01) were prepared and titrated with 1 M HCl in a borosilicate glass test tube. Solutions were titrated with 1 M HCl (4 µL per addition) via micropipette with pH monitoring and allowed to stir for 30 s after each addition of HCl before recording pH. The solutions were titrated until the pH was < 3.0. A plot of volume of acid added vs. pH was generated and formulations were compared to determine relative buffering capacities. Each sample was analyzed once and the experiment was repeated three times (N = 3) to generate a mean ± SEM.

#### 2.11.6. Osmolality and pH vs. dilution of Ketamine HCl and Captisol^®^ Formulations

Solutions of KetCap, ketamine HCl, and ketamine-HCl-Captisol^®^ (100 mg/mL ketamine FB equiv, pH 5.5 ± 0.01) were prepared and diluted to the following concentrations: 10, 25, 50 and 75 mg/mL. The pH of each concentration was measured and the osmolality of each concentration was determined. Samples were measured in triplicate and experiments were repeated three times (N = 3) to generate mean ± SEM. 

#### 2.11.7. Viscosity Measurements of Captisol^®^, KetCap, and Ketamine HCl Formulations

Solutions of Captisol^®^ (10%, 12%, and 15%), BB106 (70 mg/mL ketamine FB equiv), and ketamine HCl (70 mg/mL FB equiv) were tested for their relative viscosities using a Brookfield DV-III+ Rheometer containing a CPE-40 spindle. After establishing the appropriate gap between the sample cup and spindle, the sample (500 µL) was added to the sample cup via pipette and the following parameters were set on the instrument: Speed—100 RPM; Shear Rate—750 s^−1^; Ambient temperature. The spindle was allowed to run until the viscosity reading on the instrument stabilized (~1 min) and the viscosity was recorded. Each sample was analyzed once, and the experiment was repeated three times (N = 3) to generate a mean ± SEM.

## 3. Results and Discussion

### 3.1. Ketamine Freebase Phase Solubility with Captisol^®^


To determine the solubility of un-ionized ketamine in Captisol^®^ and, thus, the ability of the solution to solubilize un-ionized ketamine freebase, a phase solubility experiment was conducted. This also supports a 1:1 stoichiometric relationship, and allows the calculation of the stability constant (K_1:1_) for the equilibrium between ketamine (D), Captisol^®^ (CD), and the complex (D * CD) (Equation (1)). The stability constant (Equation (2)), is a measure of the affinity of the ligand for the cyclodextrin and can be used to compare guest drugs and to calculate the solubility of a given guest drug at different concentrations of cyclodextrin [3,28]. HPLC was used to quantify the concentration of ketamine in solution and a representative HPLC trace and standard curve are provided (Appendix A, respectively). A modification of a published Captisol^®^ phase solubility experiment from Ligand^24^ was used to perform the experiment. The resulting phase solubility curve is shown in Appendix A. Results showed a linear relationship consistent with a 1:1 host guest stoichiometry (A_L_ type), with the stability constant (K_1:1_) determined to be = 437.70 M^−1^ and the complexation efficacy (CE, Equation (3) below)—the ratio of complexed (D*CD) to free cyclodextrin (CD)—determined to be 1.62 [28]. This suggests that around 7 out of an available 11 Captisol^®^ molecules will be occupied by ketamine at saturating equilibrium conditions or, in other words, you need 1.6 moles of Captisol^®^ to solubilize 1 mole of ketamine freebase.
(1)$$D+CD\overset{K_{1:1}}{⇌}D*CD$$
(2)$$K_{1:1}=\frac{slope}{S_{0}\left(1-slope\right)}$$
(3)$$CE=S_{0}K_{1:1}=\frac{\left[D*CD\right]}{\left[CD\right]}$$

Loftsson et al. [28] previously evaluated the K_1:1_ and CE for 28 organic small molecule drugs with HPβCD or RMβCD (randomly methylated β-cyclodextrin). A wide range of K_1:1_ and CEs were observed. However, the average CE of these drugs was found to be ~0.3, suggesting that ~1 of every 4 cyclodextrin molecules is occupied at any time in the solution or, in other words, a 4-fold molar excess of cyclodextrin is required to solubilize the drugs. Thus, ketamine, as determined here, was found to have a relatively high CE for Captisol^®^ under the experimental conditions used. Notably, the NMDAR antagonist dextromethorphan (DXM) was reported to have a CE of 1.96 (K_1:1_ = 5900 M^−1^) with randomly methylated β-cyclodextrin (RMβCD) (suggesting two out of every three cyclodextrins are occupied) [28].

### 3.2. Preparation of Captisol Acid (CapAcid)

#### 3.2.1. Preparation of Captisol Acid (CapAcid)

To make the formulation in which Captisol^®^ acts as an anion for protonated ketamine cations, exchange of the Na^+^ of Captisol^®^ for the protonated ketamine ammonium cations had to be achieved. Several options were considered; however, the way chosen to achieve this was to synthesize the acid form of Captisol (CapAcid) using an ion exchange resin and perform an acid-base reaction with ketamine freebase. For the synthesis of CapAcid, Amberlite IR120 H-form resin was used to generate the sulfonic acid form of Captisol^®^ (CapAcid) and capture the Na^+^ cation from Captisol^®^. Both batch and column conditions were explored, though ultimately column conditions were utilized due to convenience. Dowex 50WX2 H-form (100–200 mesh) resin was also explored based on a published protocol [29], but it was found that the smaller particle size was inconvenient to utilize under our conditions. However, otherwise, the resin resulted in adequate exchange. 

Following collection of the solution from the column, the CapAcid solution can be used immediately as a solution or lyophilized and dried under vacuum to obtain the CapAcid in its solid form as a shiny, opaque, white free-flowing powder that was hygroscopic and deliquescent. Like Captisol^®^, CapAcid retained moisture, which was readily determined using Karl Fischer titration. CapAcid solid was easily handled and could be stored in the dark at −20 °C until needed. However, due to its hygroscopic and deliquescent nature, an effort was always made to limit exposure when weighing and testing moisture content, etc. If the solids were left out in ambient conditions (room temperature, fluorescent lighting, and ambient humidity), severe discoloration of the solids was observed within a few days, with clumping and puddling of the solids from high levels of moisture. No discoloration was observed after months when stored in the freezer, as described above. A NaOH titration experiment on the isolated CapAcid showed 96.6% of theory (Appendix A).

#### 3.2.2. Preparation of Ketamine Captisol^®^ Formulations

Initial work focused on a 96 mg/mL ketamine formulation in 14% (*w/v*) CapAcid (BB105). This concentration of ketamine was chosen based on several considerations, including strength, stoichiometry, and stability based on predicted solubility of un-ionized ketamine from the phase solubility experiments. The BB105 formulation confirmed the hypothesis that the KetCap solution osmolality will be substantially reduced relative to comparable ketamine HCl solutions (740.1 and 448 mOsmol/kg for 100 mg/mL ketamine HCl (N = 3) and BB105 (N = 5), respectively). This demonstrated that Captisol^®^ could be used to simultaneously raise the pH of the solution and reduce the osmolality substantially relative to a stoichiometric salt (e.g., HCl) and most non-stoichiometric salt ratios possible with common organic anions (e.g., citrate, fumarate, tartrate, etc.). However, BB105 was still hyperosmotic and ultimately an isosmotic solution was sought. Any resulting formulation also had to maintain an adequate concentration of ketamine to allow for convenient subcutaneous delivery. Too large of a volume would create delivery issues, possible irritation and packaging issues with regard to delivery devices with fixed reservoirs. To produce an adequately concentrated isotonic formulation for injection, a 70 mg/mL ketamine freebase (*w/v*) with 10% (*w/v*) CapAcid was selected based on osmolality vs. dilutions of BB105 (data not shown) and estimates from predicted osmolarity calculations. The 70 mg/mL ketamine freebase concentration would still provide a concentrated solution of ketamine, minimizing the overall volume delivered to patients, and be able to fit within the restricted storage capacity of an SC delivery device. The 70 mg/mL formulation (BB106) was subsequently prepared and found to be isosmotic (Table 1), with a mean osmolality of 291.4 ± 4.0 mOsmol/kg, (N = 7, 95% CI, 283.5–299.2 mOsmol/kg). By comparison, isotonic 0.9% saline was found to be 290.5 ± 2.88 mOsmol/kg, (N = 13, 95% CI = 287.2–294.8 mOsmol/kg) and 70 mg/mL ketamine HCl (FB equiv), 527.0 mOsmol/kg. The density of BB106 was measured by mass (N = 3) and determined to be 1.0505 ± 0.004 g/mL. Finally, to demonstrate the feasibility with ketamine enantiomers, both EsketCap (BB107) and ArketCap (BB108) were prepared at 70 mg/mL (using esketamine or arketamine free base, respectively) resulting in an average osmolality of 298.2 mOsmol/kg (N = 2) for BB107 and 289.3 mOsm/kg (N = 1) for BB108. BB107 has the advantage of dramatically increasing the potency of the formulation, as esketamine is almost 2× the potency of the racemate as an analgesic and antidepressant [30,31,32]. R-ketamine has also shown potential in preclinical models as a therapeutic for neuropsychiatric indications including depression [33,34]. 

As comparators, Captisol^®^ formulations with ketamine HCl (Ketamine HCl Captisol^®^) were prepared in addition to another common, but non-ionic, cyclodextrin, hydroxypropyl-β-cyclodextrin (HPβCD). A propylene glycol formulation was also evaluated as it is a commonly used as a cosolvent, used to enhance solubility of APIs. Results are presented in Table 1. As stated previously, BB106, BB107, and BB108 are isosmotic and comparable to 0.9% saline. In contrast, the resulting osmolalities of ketamine-HCl-Captisol^®^ formulations were notably hyperosmotic (>900 mOsm/kg, Table 1), exceeding that of ketamine HCl. High osmolalities were also observed with the propylene glycol and HPβCD formulations. The osmolalities of KetCap, ketamine HCl, and Ketamine-HCl-Captisol^®^ at different concentrations of ketamine (prepared via serial dilution) were determined and are presented in Figure 2. The osmolality of KetCap was lower at every concentration relative to ketamine HCl, and ketamine HCl Captisol^®^. The results in Figure 2 illustrate the dramatic difference in osmolality obtained by using Captisol^®^ as the anion in the formulation (as in BB106) versus just mixing ketamine HCl with Captisol^®^.

### 3.3. Analysis and Stability of Ketamine-Captisol^®^ Formulations Compared to Ketamine HCl

#### 3.3.1. Determination of Concentration of Ketamine in KetCap and Commercial Ketamine HCl Samples

The concentration of ketamine in BB106 was determined by HPLC along with the concentrations of ketamine (FB equiv) in various brands of commercial ketamine HCl samples (50 mg/mL and 100 mg/mL). Concentrations are presented in Table 2. 

#### 3.3.2. Buffering Capacity of Ketamine Captisol^®^ Complexes Compared to Ketamine HCl

As mentioned previously, one advantage to using Captisol^®^ in the KetCap formulations is the ability to raise and stabilize the pH of the ketamine solution relative to commercial HCl formulations. Stability to changes in pH (buffering capacity) of a formulation is important for product shelf-life as well as pharmacological properties such as absorption from the site of administration [9]. As such, the buffering capacity of KetCap formulations to added acid or base was evaluated relative to comparator formulations. 

Titration of each solution with 2 M NaOH found varying ranges of tolerability to added base before precipitation, as summarized in Table 3. As expected, the KetCap and Captisol^®^ containing formulations were able to withstand more added base, since any un-ionized ketamine freebase formed could be complexed and solubilized by the cyclodextrin. Interestingly, BB106 showed increased buffering capacity relative to ketamine-HCl-10%-Captisol^®^ and ketamine HCl hydroxypropyl-β-cyclodextrin (HPβCD). Ketamine HCl and 10% propylene glycol had the least ability to withstand added base before precipitation. The high buffering capacity of BB106 against added base is important in that it has the potential to ensure stability of the formulation to environmental conditions but also may contribute to buffering at the site of administration when the pH would be anticipated to be raised relative to the pH of the formulation.

The stability of the formulations to added acid (1M HCl) was also evaluated (Figure 3). The pH of the ketamine HCl solution dropped steeply upon addition of HCl. As expected, BB106 showed greater resistance to added HCl compared to the ketamine HCl solution. Interestingly, the ketamine-HCl-Captisol^®^ formulation appeared to exhibit superior acid buffering capacity to BB106, though the magnitude of the effect is minor and this may be due to the difference in ionic strength. Notably, this is the opposite of the trend seen with added base. Regardless, in the presence of Captisol^®^, the formulations resisted pH changes under added acid or base conditions in a manner superior to ketamine HCl, suggesting that the KetCap formulations prepared are stable with respect to pH under appropriate storage conditions. This interesting buffering property observed for KetCap formulations is presumed to result from the potential of un-ionized ketamine freebase (guest) to complex to the Captisol^®^ (host) (Figure 1). In the case of added base, any un-ionized ketamine formed is able to complex to uncomplexed Captisol^®^ present in the solution, solubilizing it, while at the same time removing the un-ionized ketamine species from solution while minimizing it as a contributor to the solution pH. Why complexed ketamine freebase seems to be removed from the pH equilibrium is an interesting question for future work. In the case of added acid, un-ionized ketamine freebase that is present in solution in a largely Captisol^®^-complexed form, will become protonated (and presumably un-complex, albeit this is a simplification and equilibrium effects will govern the process), neutralizing the added acid and, thus maintaining the pH of the solution. The buffering capacity can be modulated by exploiting the concentrations of the relevant species. 

The relationship between ketamine concentration and pH was determined for commercial pH mimicking Ketamine HCl, BB106, and Ketamine-HCl-Captisol^®^. Based on acid-base theory, it was expected that the pH would rise with decreasing concentration of the protonated ammonium ketamine species, since the H^+^ concentration is decreased. Results are shown in Figure 4 and were consistent with this expectation. Ketamine HCl had a lower pH (more acidic) at every comparable concentration relative to BB106 and ketamine-HCl-Captisol^®^ formulations. 

### 3.4. Stability of Captisol^®^


#### Fehling’s Test for Stability of Captisol^®^

As Captisol^®^ is a cyclic polysaccharide, with its sugar subunits linked by α-(1,4) glycosidic bonds, it was necessary to assess its stability to acid hydrolysis. Thus, stability studies were performed on Captisol^®^ to assess its stability to acidic process conditions using the Fehling test, a qualitative colorimetric assay for the presence of reducing sugars. In this test, the blue, copper (II) reagent reacts readily with reducing sugars to form a red, copper (I) complex that precipitates and a color change to the solution (Appendix A). Therefore, if Captisol^®^ were to decompose to a significant extent through acid hydrolysis of its glycosidic bonds, during the preparation of the acid form, a positive Fehling test result would be obtained. Control experiments were performed to assess assay sensitivity, with 2 mg/mL glucose being reliably detected under these conditions (Appendix A). Likewise, it was confirmed that 2 mg/mL glucose could be detected in the presence of Captisol^®^. 

The stability of freshly prepared CapAcid was assessed over time starting from 15% and 20% Captisol^®^ solutions. We initially studied solutions of CapAcid prepared from 15% Captisol^®^ via the ion exchange procedure described above. The solution collected from the ion exchange column was aliquoted into test tubes for the Fehling tests and two small aliquots were lyophilized to determine the concentration of the CapAcid solution by mass (11.4 ± 0.2%). All solutions tested negative for decomposition up to 72 h stored in the dark at room temperature (Appendix A).

CapAcid solutions prepared from 20% Captisol^®^ were also tested. The concentration of CapAcid was determined by mass (15.4 ± 0.08 %) and tested for stability up to 168 h. All solutions of CapAcid tested negative for decomposition up to 48 h (Appendix A). It should be noted that a small amount of red precipitate was observed in samples at 48 h when cleaning the test tubes after the Fehling test, but visual observations during the test in all cases indicated a negative result. CapAcid showed a positive result for hydrolysis with the 168 h sample (Appendix A). Interestingly, solid CapAcid gave weak positive results, suggesting some hydrolysis occurs during the lyophilization process. However, this was not further evaluated and more quantitative methods are ultimately needed. The results are consistent with the fact that β-cyclodextrins are known to have greater stability to acid hydrolysis than non-cyclic linear counterparts due to the non-terminal glycosidic bonds [35].

### 3.5. Viscosity Measurements of Captisol^®^, KetCap, and Ketamine HCl Formulations

Viscosity can influence the injectability of a parenteral formulation including ergonomics, dose preparation, dose administration and perceived injection site pain [36,37]. The subcutaneous injection (bolus) of higher viscosity solutions (15–20 cP) was found to be better tolerated than lower solutions (1, and 8–10 cP) [36]. The extent to which viscosity influences pain with a slow infusion was not assessed here. Higher viscosity formulations often present additional challenges for administration [37]. Since this formulation was designed to be used for subcutaneous injection, the relative viscosities of BB106 and a comparable concentration of ketamine HCl were determined. Captisol^®^ was also measured for comparison at three different concentrations. The results are shown in Table 4. The viscosity of ketamine HCl (70 mg/mL) was close to that of pure water (Table 4). While we did not measure commercial ketamine samples, we anticipate them to be close to the 70 mg/mL ketamine HCl sample and lower than BB106. KetCap (BB106) had a slightly raised viscosity (1.86 ± 0.04) comparable to the 15% Captisol^®^ solutions. These results confirm that BB106 has acceptable viscosity for subcutaneous delivery. Preliminary experiments in 28-gauge insulin needles and V-GO^®^ insulin pumps (Valeritas Inc, Marlborough, MA, USA) show favorable flow characteristics.

## 4. Patents

This work has been granted a United States Patent; US Patent No. 10,973,780.

## Figures and Tables

**Figure 1 pharmaceutics-14-00556-f001:**
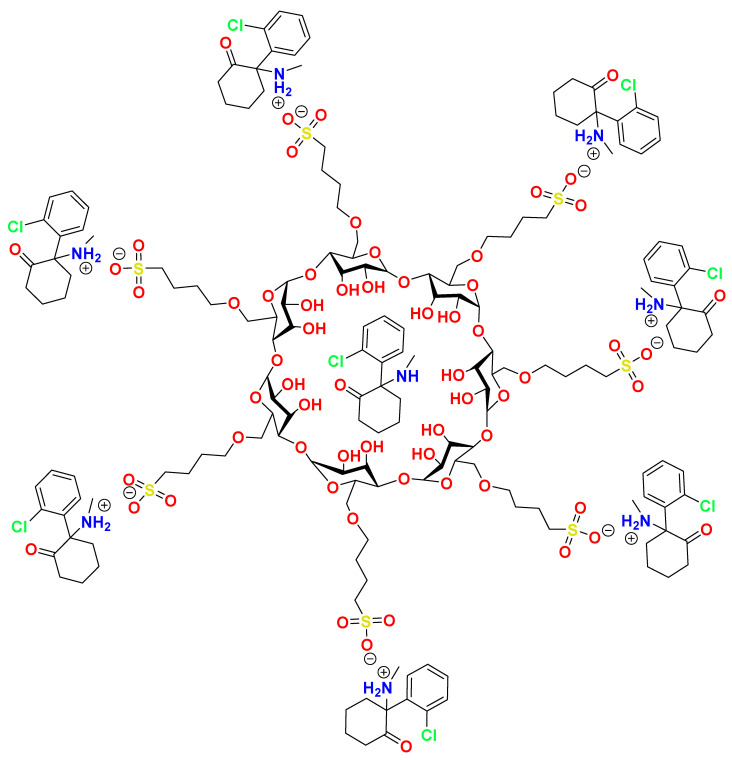
Proposed salt formation and complexation of ketamine with Captisol^®^ to form KetCap.

**Figure 2 pharmaceutics-14-00556-f002:**
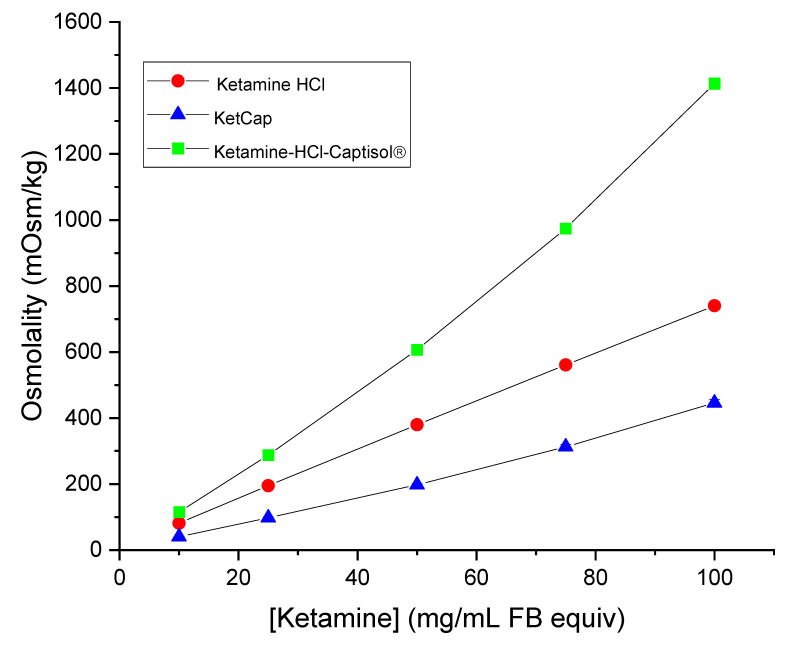
Dilution vs. osmolality of Ketamine HCl and formulations. Measured osmolalities of Ketamine HCl, KetCap, and Ketamine-HCl-Captisol^®^ at different concentrations obtained following dilution of 100 mg/mL stock formulations. Data presented as mean ± SEM. N = 3.

**Figure 3 pharmaceutics-14-00556-f003:**
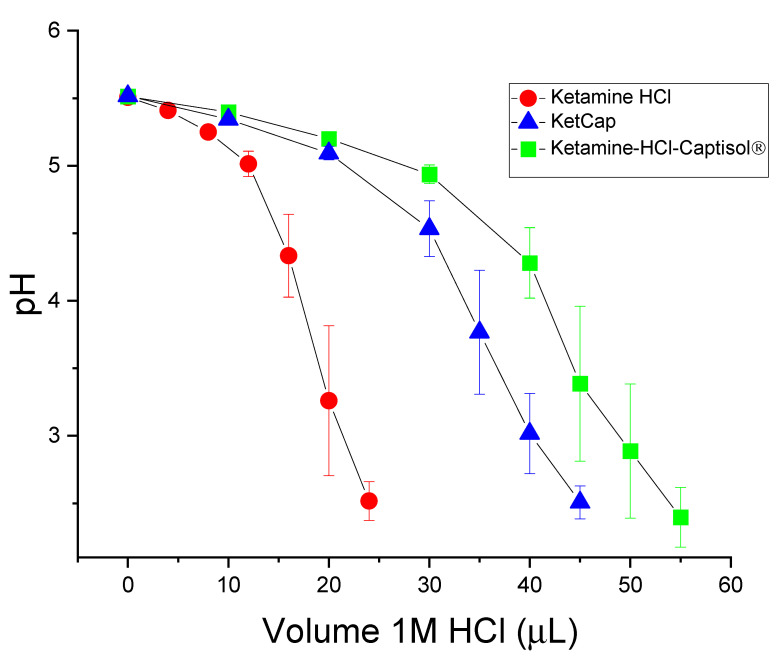
Titration of 100 mg/mL Ketamine, KetCap, and Ketamine-HCl-Captisol^®^ Formulations with 1M HCl. Starting pHs 5.50 ± 0.01. Experiments carried out on a 2 mL volume. Data presented as mean ± SEM. N = 3.

**Figure 4 pharmaceutics-14-00556-f004:**
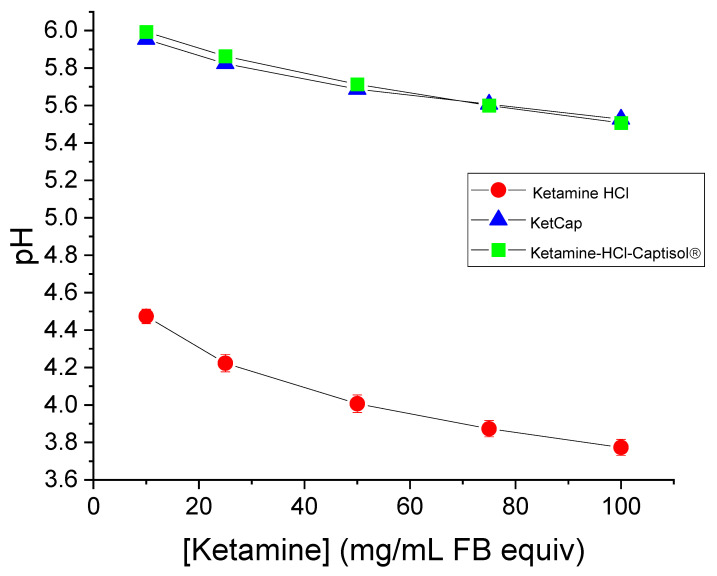
Relationship of ketamine concentration on pH, for Ketamine HCl, KetCap, and Ketamine-HCl-Captisol^®^ formulations. Data presented as mean ± SEM. N = 3.

**Table 1 pharmaceutics-14-00556-t001:** Average osmolality of Ketamine HCl with and without additives and Ketamine Captisol^®^ formulations.

Sample	[Ketamine] (mg/mL FB eq)	Average Osmolality (mOsm/kg) ± SEM
0.9% Saline	-	290.8 ± 1.1
Ketamine HCl	70	527.0 ± 2.7
Ketamine HCl	96	704.4 ± 9.9
Ketamine HCl	100	740.1 ± 6.8
BB105 (KetCap)	96	448.0 ± 4.4
BB106 (KetCap)	70	291.4 ± 4.0
BB107 (EsKetCap)	70	298.2 ± 9.5 ^+^
BB108 (ArKetCap)	70	289.3 *
Ketamine-HCl-10.7%-Captisol^®^	70	938.5 ± 2.2
Ketamine-HCl-15%-Captisol^®^	96	1443.4 ± 28.0
Ketamine-HCl-10%-HPβCD	70	618.0 ± 5.7
Ketamine-HCl-10%-Propylene glycol ^1^	70	1899.0 ± 49

^1^ Sample diluted 2:1 in HPLC water for osmolality reading and osmolality was back-calculated from this dilution factor. ^+^ N = 2. * N = 1.

**Table 2 pharmaceutics-14-00556-t002:** Concentration of Ketamine in BB106 and commercial Ketamine HCl samples. Commercial ketamine HCl samples (Ketalar, Mylan, Hospira, and West-Ward) were sampled from two separate bottles, each measured in triplicate.

Sample	Theoretical [Ketamine] (mg/mL FB eq)	[Ketamine] (mg/mL FB eq) ± SEM	Measured pH (Mean ± SEM)
BB106	70	73.1 ± 1.3	5.51
BB106 10-month stability	70	72.4 ± 0.4	-
Ketalar Ketamine HCl	50	52.8 ± 0.2	4.09 ± 0.01
Mylan Ketamine HCl	50	51.7 ± 1.1	4.18 ± 0.03
Hospira Ketamine HCl	100	103.8 ± 2.0	3.88 ± 0.03
West-Ward Ketamine HCl	100	105.9 ± 3.0	3.94 ± 0.02

**Table 3 pharmaceutics-14-00556-t003:** Forced precipitation of Ketamine with 2M NaOH after pH 5.5, starting pHs 5.50 ± 0.01. Experiments performed on a 2 mL volume. NaOH added in 1 μL volumes. Data presented as mean ± SEM. N = 3.

Sample	Mean Volume NaOH to Precipitate (µL) ± SEM
Ketamine HCl (70 mg/mL FB equiv)	5.33 ± 0.33
Ketamine-HCl-10%-HPβCD (70 mg/mL Ketamine FB equiv)	7.67 ± 0.33
Ketamine-HCl-10%-Propylene glycol (70 mg/mL FB equiv)	5.33 ± 0.33
Ketamine-HCl-10%-Captisol^®^ (70 mg/mL Ketamine FB equiv)	8.33 ± 0.67
BB106 (70 mg/mL Ketamine FB equiv)	9.67 ± 0.67

**Table 4 pharmaceutics-14-00556-t004:** Measured viscosities of Captisol^®^, Ketamine HCl, and KetCap Solutions. Samples were freshly prepared and measured on separate days to generate a mean ± SEM. N = 3.

Sample	Mean Viscosity (cP) ± SEM
H_2_O	1.01 ± 0.03
Ketamine HCl (70 mg/mL FB eq)	1.19 ± 0.04
BB106 (70 mg/mL Ketamine)	1.86 ± 0.04
10% Captisol^®^	1.58 ± 0.02
12% Captisol^®^	1.63 ± 0.03
15% Captisol^®^	1.75 ± 0.02

## Data Availability

Data is available in the text, Appendix A or available upon request from the corresponding author.

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
