# Peer review of "Three Birds, One Excipient: Development of an Improved pH, Isotonic, and Buffered Ketamine Formulation for Subcutaneous Injection"

_pharmaceutics, 2022, doi:10.3390/pharmaceutics14030556_

Round 1

Reviewer 1 Report

In the present article, authors describe a new formulation for Ketamine that can be applied for subcutaneous injection. The manuscript is well written even if some parts of the Material and Methods section can be shortened. All experiments are carefully designed and performed and represented in great detail. Thus, I would like to recommend the publication of this manuscript in Pharmaceutics after the following issues are addressed.

Minor issues:

Introduction:

lines 23-24: “As such, various cyclodextrins have found use in consumer, industrial and medicinal products” Provide citations here.

lines 27-28: ”The resulting complex can be referred to as a supramolecular guest-host interaction”. Rephrase for clarity.

lines 28-30: ”The resulting …cyclodextrin host.” Are there any citations here?

Materials and Methods

lines 358-359: “2.11.4. Preparation of Ketamine HCl-Captisol® Formulation (70 mg/mL Ketamine HCl 358 FB Eq)”. Renumber paragraph.

Results and Discussion

line 499, Par. 3.1.2. Authors should also state the type of cyclodextrin complex (i.e. AL type) that shows 1:1 host guest stoichiometry.

Author Response

Thank you for you input. The requested corrections have been incorporated. 

Reviewer 2 Report

Gamrat and co-workers have studied the novel idea of Ketamine Formulations for SC Injection using Captisol as main excipient. The work is of high significance to the community. However, there are few things that needs further clarification:

1) Please provide reference for the statement to support the claim "It is probable these adverse reactions are due, at least part, to a low pH and/or high osmolality characteristic of commercial ketamine HCl formulations."

2) Please make it more clear for the purpose of CapAcid? Why it is being prepared and its uses, compared to Parent excipient Captisol?

3) Please provide proper justification for this statement in section 2.6 phase solubility studies in methods section "sample was diluted in 7:3 formate:acetonitrile buffer with 20 μg/mL procaine HCl to an estimated target range of 20-40 μg/mL." And its impact on the phase solubility studies.

4) Please provide any reference for the HPLC sample preparation and analysis ! If the method is in-house developed method, please confirm the validation of the method.

5) Please provide viscosity of the current marketed IV/SC formulations of ketamine for comparison sake would be helpful for the readers.

Author Response

Thank you for your input. The corrections have been incorporated. 

  1. Added language to state this is our hypothesis based on literature cited.
  2. Under Section 3.2.1 we better explain the usage of CapAcid in order to allow the use of Captisol as the anion for protonated ketamine ammonium cations.
  3. The statement has been clarified to make it clear that this procedure is part of the analysis of ketamine concentration in the solutions. 
  4. References and clarification statements have been added to Section 2.3.6 HPLC.
  5. We agree this would be useful information. Unfortunately we do not have access to the equipment at the present time. We did however add some language that we anticipate the values would be fairly close to the 70 mg/mL ketamine HCl comparator solution (which itself had a fairly low viscosity) used in the experiment and below that of BB106.